# Economic Growth, Income Inequality and Food Safety Risk

**DOI:** 10.3390/foods12163066

**Published:** 2023-08-15

**Authors:** Yong-Qi Chen, You-Hua Chen

**Affiliations:** 1Collage of Economics and Management, South China Agricultural University, Guangzhou 510642, China; cyq99@stu.scau.edu.cn; 2Research Center for Green Development of Agriculture, South China Agricultural University, Guangzhou 510642, China

**Keywords:** food safety risk, economic growth, income inequality, Kuznets curve, moderating effect

## Abstract

Food safety risk, as an implicit cost of social and economic development, endangers the health of global residents, including China. To systematically understand the impact of socioeconomic development on food safety risk and to establish a sound modern governance system of food safety in China, this paper uses provincial panel data from 2011 to 2020 to explore the relationship between food safety risk and socio-economic development factors such as economic growth and income inequality by employing a two-way fixed effect model and moderating effect model. The results show that the food safety risk is a Kuznets curve, and the turning point is about RMB 58,104.59 per capita GDP (based on prices in 2011). However, under the moderating effect of income inequality, the turning point of the Kuznets curve of food safety risk will shift to the right, and the curve will be flattened. In other words, income inequality has a negative moderating effect on the “inverted U-shaped” relationship between economic growth and food safety risk. When dealing with food safety problems, the goal of stable and sustained economic growth and common prosperity should be incorporated into policy formulation to enhance the governance effectiveness of food safety risk.

## 1. Introduction

Food safety is one of the world’s most important and common public challenges. Globally, 600 million people (nearly 1 in 10) fall ill, and 420,000 die each year from contaminated food (who.int, accessed on 30 April 2020). The situation in China is also serious. According to the Global Food Safety Index Report 2021, China ranks 34th in the food safety index among 113 countries. In 2020, there were as many as 7073 outbreaks of foodborne diseases in China (China Health Statistics Yearbook 2021). Food safety has become China’s second biggest public concern [1]. To alleviate public anxiety and ensure people eat healthily, the Communist Party of China has attached great importance to food safety. It has incorporated food safety issues into major national strategies since the 18th National Congress of the Communist Party of China.

Food safety is essentially an economic issue [2]. It is the implicit cost of economic growth, while governance also depends on the support of economic growth, so it is closely related to the level of economic growth [3,4,5]. Olkiewicz and Wolniak believe there is a positive correlation between per capita GDP and food safety through the analysis of EU countries [6]. However, food safety risks mainly come from chemical, microbial, and physical hazards [7,8], and different hazards are alleviated or highlighted with economic growth. There should be a non-linear relationship between economic growth and food safety risk [9,10], and this non-linear relationship may be an “inverted U-shaped” relationship [2]. Accordingly, Zhang and Sun point out that food safety risk is a quasi-Kuznets curve [3]. Subsequently, Zhang et al. and Yin et al. confirmed the “inverted U-shaped” relationship between economic growth and food safety risk. They proposed the food safety Kuznets curve hypothesis through econometric analysis of China’s food safety situation [11,12]. However, Zhang et al. find that the “inverted U-shaped” relationship between economic growth and food safety risk is not necessarily statistically significant [13]. Nowadays, Scholars have different views on the relationship between economic growth and food safety risk, and there is a relative lack of complete and systematic theoretical analysis.

In addition, Income inequality reflects differences in the public’s living conditions and risk exposure [14]. Scholars mostly analyzing the relationship between economic growth and food security risks base their analysis on economic growth itself, ignoring the impact of income inequality on the relationship between economic growth and food safety risk. According to the Easterlin Paradox, factors such as inequality offset the positive effects of economic growth to varying degrees. The increase in income inequality, as a deep-seated cause of China’s food safety problems [15], will undermine social order and trust [16,17,18], deepen the differences in geographical isolation [17,19,20], innovative technology adoption and education [20,21,22], and then lead to differences in environment and food access. This increases the level of food safety risk exposure among the middle and low-income groups [23], thereby increasing society’s overall food safety risk at the same level of economic growth. Therefore, it may be biased to directly analysis the relationship between economic growth and food safety risk by ignoring the moderating effect of income inequality.

A systematic understanding of the impact of socioeconomic development on food safety risk will not only help enrich the theory of food safety governance; it can also improve people’s awareness, prevention, and management of food safety, which has important practical significance. Based on the existing analysis, and considering the moderating effect of income inequality, this paper analyzes the impact of economic growth on food safety risk from the causes and mechanism of food safety risk. And we also focus on the moderating effect of income inequality on the relationship between economic growth and food safety risk. The results show that the food safety risk in China is a Kuznets curve, and the turning point is near RMB 58,104.59 per capita GDP (based on prices in 2011). While under the moderation of income inequality, the turning point of the Kuznets curve of food safety risk will shift to the right, and the curve will become flatter.

The contribution of this paper is twofold. Firstly, our study improves the mechanism of economic growth on food safety risk and further confirms the Kuznets curve hypothesis of food safety risk. Secondly, our study reveals the moderating effect of income inequality on the non-linear relationship between economic growth and food security risk. It further reveals the impact of socioeconomic development on food safety risk, supplements the knowledge of food safety risk and its governance, and provides a new perspective of policy-making in China to improve food safety governance and ensure food safety for residents.

## 2. Theoretical Analysis and Hypotheses

### 2.1. Economic Growth and Food Safety Risk

From the Food and Agriculture Organization of the United Nations, World Health Organization, it learns that food safety is the absence of acute or chronic hazards to consumers when food is prepared and consumed as intended. Risk is defined as “the effect of uncertainty on objectives” by ISO 31000. In the field of food, risk is generally defined as a combination of the likelihood that a hazard may occur in the food and the magnitude of the effect of the exposure hazard on human health (Codex Alimentarius 2001). This means that the risk is a function of the presence of the hazard and the severity of its impact on human health [24]. Accordingly, food safety risk is defined as the possibility and severity of adverse effects on human health caused by hazards in food (Codex Alimentarius Commission), including physical, chemical, and microbiological hazards [7,8]. These are alleviated and aggravated correspondingly at each level of economic growth. Therefore, this paper analyzes the relationship between economic growth and food safety risk based on the composition of food safety risk and economic principles.

The level of food safety risk is lower during the low economic growth stage. At this time, as the leading industry of the national economy, agriculture has sufficient land and labor input, relatively fertile and unpolluted soil, less input of pesticides, fertilizers, veterinary drugs, and other chemicals [25], and the problem of pollution at the source of food production is not serious. In addition, the food supply chain is shorter, the degree of processing is lower, and the demand for transportation is lower, so the probability of contamination in the supply chain is also lower. At this stage, the main food safety risks mostly occur in household consumption and storage. Because the refrigeration technology is not extensive and the cooking conditions are limited, it is easy to produce microbial hazards such as mildew.

When an economy enters the rapid growth stage, with the rapid advancement of industrialization and urbanization, food safety problems emerge constantly. Food safety risks increase with the improvement of economic growth level. Firstly, Due to the extrusion of cultivated land area, labor force and capital by industrialization and urbanization, a large number of chemical materials such as pesticides, fertilizers and veterinary drugs are used in agricultural production to increase production [11,26,27], which increases the chemical pollution at the source of food production. At the same time, the discharge of industrial waste gas, wastewater and other wastes leads to the pollution of the environment, water and soil, resulting in a large number of microbial pathogens and heavy metals and other chemical pollutants, which directly threatens the drinking water safety of residents and causes the source pollution of agricultural, forestry, animal husbandry and fishery products [28,29]. In addition, with the division of labor and diversified demand brought by industrialization and urbanization, the food supply chain’s extension and division of labor will produce serious information asymmetry problems at this stage, leading to moral hazards and a series of food safety risks.

Moreover, in the processing and packaging process, the overall cultural and technological level of society is low, and the food industry has not yet formed unified processing and packaging standards, leading to food’s physical and microbial risks in the processing and packaging environment. In the process of storage and transportation, the lag and incomplete equipment of cold chain technology increases the microbial risk of food. In the consumption process, the emergence of ready-to-eat food and street sales increases the microbial risk of food in the absence of unified supervision, refrigerators and tap water [30,31,32]. In particular, street food is also exposed to pollutants such as dust and flies, which increases the transmission of bacteria and viruses in food [32]. In this phase, food safety’s physical, chemical, and microbial risks have erupted together, and food safety problems are becoming increasingly serious.

When the economy enters the developed stage, the environment and food safety management are constantly improving, and the food safety problem has been alleviated to a certain extent and entered the declining stage under the sustainable development of high quality. Firstly, food safety risks at the source of production, the chemical and microbial risks at the source of food production have been reduced due to the improvement and control of chemical materials input into production, the monitoring and control of industrial waste discharge, and the improvement of environmental and water pollution. Secondly, although the food supply chain has been further extended and refined, the moral hazard caused by information asymmetry has been reduced because of the popularization of traceability systems, information technology and cold chain technology, and the improvement and supervision of safety standards in all links. This improves food safety throughout the supply chain [8]. Moreover, with the overall improvement in education level, the effect of moral culture and knowledge technology began to appear. Therefore, production and processing personnel and consumers have formed self-restraint and mutual supervision [32,33]. Food safety issues have been taken seriously and managed. At this stage, food safety risks are mainly reflected in microbial risks, mostly in the final consumption stage, such as restaurants and families. However, With the improvement of food safety awareness and the improvement and popularization of technology, the risk of food microorganisms has been improved to a certain extent, and the overall risk of food safety has shown a downward trend.

In summary, the initial stage of pursuing rapid economic growth has brought a serious gap between supply and demand and ecological environment pollution. Technology and resource constraints result in a large input of chemical materials and microbial contamination. At this time, physical, chemical, and microbiological risks are emerging together; food safety risks are rising [32,34,35,36]. However, as the economy grows, the effects of industrial structure and technology appear. People’s requirements, awareness, and ability to pay for food safety improve, forcing enterprises and the government to pay attention to the production and supervision of food safety. Food physical, chemical and microbial risks have been alleviated to a certain extent. Food safety risks have begun to decline [13,32,33]. Based on the above analysis, this paper proposes Hypothesis 1 (H1).

**Hypothesis** **1** **(H1).***There is an “inverted U-shaped” relationship between economic growth and food safety risk*.

### 2.2. The Moderating Effect of Income Inequality

Income inequality can affect the impact of economic growth on food safety risks [37]. It not only increases the food safety risk exposure of the low and middle-income group [23] but also increases the food safety risk of the whole society by causing environmental degradation [38,39] and inhibiting technological innovation [40].

Income inequality differentiates risk exposure [14]. Under the same economic level, the food safety risks for the low and middle-income groups will be higher [23,30]. Low- and middle-income groups are deprived of better food consumption markets as rising income inequality exacerbates geographic isolation [17,19,20]. Their relatively poor market environment and technical facilities [19,20], weak government regulation, limited resources, and small and untrained food supply staff [18,26,30] make them more vulnerable to some chemical and microbial food hazards. In addition, the increase in income inequality reduces the income distribution of low and middle-income groups. In the face of rising food prices brought about by economic growth, their real income may be reduced. Therefore, price-sensitive consumers generally choose cheap and low-quality food based on the principle of “price” priority [3,23], which provides a consumer market for low-quality and substandard food.

Moreover, their education level is relatively low. They have less knowledge, treatment, and guidance on food safety [8,41,42], providing a market for some contaminated foods that do not meet market standards [32,43]. Finally, growing income inequality creates more economic uncertainty for low and middle-income groups, making them make difficult trade-offs between basic needs and long-term benefits, increasing the difficulty and anxiety in their food preparation [23], which increases the physical and microbiological risks of food purchase, cooking and storage at home. The low and middle-income groups will suffer more unsafe food hazards, thus changing the shape and inflection point of the Kuznets curve of food safety risk.

The increase in income inequality will lead to environmental pollution and the lag of technological innovation, which increases the probability of environmental pollution of food in the supply chain. Firstly, environmental pollution is the second largest risk factor causing food safety problems [42]. Income inequality will aggravate environmental pollution [38] and delay the emergence of the inflection point of the environmental Kuznets curve [39]. Thus, it indirectly increases the overall level of food safety in society and moves the inflection point of the Kuznets curve of food safety risk to the right. Secondly, technology is essential for food storage, transportation, and supervision. The inhibition of income inequality on technological innovation and upgrading [40,44] will increase the chemical risk of food in production and processing and the microbial risk in storage and transportation, making the Kuznets curve of food safety risk becomes flatter. Accordingly, this paper puts forward the following hypotheses:

**Hypothesis** **2** **(H2).***Income inequality moderates the inverted U-shaped relationship between economic growth and food security risk*.

**Hypothesis** **2a** **(H2a).***Under the moderating effect of income inequality, the turning point of the “inverted U-shaped” curve of economic growth and food safety risk shifts to the right*.

**Hypothesis** **2b** **(H2b).***Under the regulation of income inequality, the “inverted U-shaped” curve of economic growth and food safety risk is flatter*.

## 3. Materials and Methods

### 3.1. Data

We use data from the China Health Statistics Yearbook (CHSY), the China Statistical Yearbook, the China Food Industry Yearbook, and National Meteorological Science Data. Where panel data for 30 provincial-level administrative regions of China (27 provinces and four municipalities directly under the central government in China mainland, but excluding Tibet, Hong Kong, Macao, and Taiwan) from 2011 to 2020. Statistics of foodborne disease events in the CHSY started in 2011.

The China Health Statistics Yearbook provides us with data on the number of foodborne disease events, patients, and personnel in health supervision institutes (centers). The China Statistical Yearbook gives us data on GDP per capita, the total output value of agriculture, forestry, animal husbandry and fishery, consumer price index, disposable income of urban and rural residents, urban and rural population, total population, and the education level (illiterate, primary school, junior high school, secondary vocational school, senior high school, junior college, university and graduate students) of people above six years old. The China Food Industry Yearbook offer us the total output value of food enterprises above scale. The National Meteorological Science Data (dataset of daily surface climatological data over China V3.0) offered us details about temperature and rainfall.

### 3.2. Variables

(1)Dependent variable: Food safety risk. Based on the main manifestations of food safety risk, we selected the number of foodborne disease events as a proxy variable for food safety risk [10].(2)Independent variable: Economic growth. According to Zhang et al. and Yin et al. [11,12], GDP per capita was selected as a measure of the level of economic growth. To eliminate the impact of price changes, GDP per capita was calculated at the constant price converted from the base period price in 2011.(3)Moderator. Income inequality. The Theil and the Gini are the best indicators to measure income inequality in different years or regions [45]. Moreover, compared with the Gini, the Theil can reflect the intra-group gap and describe the inter-group gap. And it is more sensitive to the change of the lowest income group and the highest income group, which is more in line with the actual situation in China [46]. Therefore, this paper uses Theil as the core measure of income inequality.


(1)
Theil=∑i=12(ViV×ln(ViVPiP))


In Equation (1), V denotes the total income of urban and rural areas, $V_{i}$ denotes the total income of urban or rural areas, P denotes the total population of urban and rural areas, $P_{i}$ denotes the population of urban or rural areas, i=1 denotes urban areas, i=2 denotes rural areas.

(4)Control variables: Following Zhang et al., Yin et al. and Zhang et al. [11,12,13], we control the government regulation (the number of personnel in health supervision institutes (centers)), industrial structure (the output value of tertiary industry/the output value of secondary industry), food industry output value (FIV: the total output value of food enterprises above the scale), the total output value of agriculture, forestry, animal husbandry and fishery (AFTV), Average Education level (Education level = 0 × illiterate + 6 × Number of primary school students + 9 × Number of junior high school students + 12 × Number of secondary vocational school students + 12 × Number of senior high school students + 15 × Number of junior college students + 16 × Number of university students + 19 × Number of graduate students), consumer price index (CPI), temperature and rainfall. In addition, we include providing and year fixed effects.

To eliminate the heteroscedasticity, we take the logarithm of the variables. The specific definitions and descriptions of the variables are shown in Table 1.

In Figure 1, we fit a graph of the relationship between economic growth and food safety risk. We find an ‘inverted U-shaped’ relationship between economic growth and food safety risk, a Kuznets curve.

In Table 2, we group the food safety risk according to the 25% and 75% quantiles of economic growth and the median of Theil. According to the longitudinal comparison of the mean value of food safety risk in each group, it is found that there is an “inverted U-shaped” relationship between economic growth and food safety risk in the Low-Theil group. In the High-Theil group, the relationship between economic growth and food safety risk is positive, which has not yet reached the turning point. In a cross-sectional comparison, the curve between economic growth and food safety risk is flatter in the High-Theil group than in the Low-Theil group.

### 3.3. Econometric Model

The Kuznets curve is the curve hypothesis of the American economist Kuznets in analyzing the change of income distribution status with the process of economic development [47]. Based on changing the industrial structure from traditional agricultural to modern industry, Kuznets depicted the nonlinear relationship of income inequality rising and declining with economic growth and proposed the Kuznets curve theory. Subsequently, scholars found that socio-economic-ecological factors such as secondary industry and environmental pollution also showed a non-linear relationship with economic growth, so the Kuznets curve theory has been widely used and updated in socioeconomic, environmental, and other socio-economic-ecological fields. As a prominent problem in socio-economic development, food safety risk shows a nonlinear trend of rising and falling with the industrial structure change from agriculture to industry and urbanization according to observation and data fitting, which is highly fitting with the analysis of the Kuznets curve theory. Therefore, this paper puts forward the Kuznets curve hypothesis of food safety risk and verifies it using the following econometric model.

#### 3.3.1. Fixed-Effect Model

According to the Hausman test: *p* = 0.0147 < 5%, we use the fixed effect model to identify the nonlinear impact of economic growth on food safety risk. The basic model is shown in Equation (2):(2)$$Y_{it}=\alpha +\beta_{1}X_{\;\mathrm{it}}+\beta_{2}{\left(X_{it}\right)}^{2}+\beta_{k}Z_{k\;it}+\mu_{i}+\delta_{i}+\epsilon_{it}$$
where $Y_{it}$ denote the food safety risk of the provide i in year t. $X_{it}$ denote the economic growth of the provide i in year t. Z is a vector of the control variables listed in the previous section. α is a constant term. β is the corresponding coefficient vector, while $\beta_{1}$ and $\beta_{2}$ are the two parameters we are interested in. $\mu_{i}$ is the provide fixed effects, $\delta_{i}$ is the year fixed effects, $\epsilon_{it}$ an error term and is assumed to be normally distributed.

#### 3.3.2. Moderating Effect Model

To test the moderating effect of income inequality on the “non-linear relationship between economic growth and food safety risk”, we refer to the model design of Wen et al. and Haans et al. [48,49] and introduce income inequality as a moderator based on the curvilinear regression results of Equation (2). The basic model is as follows:(3)$$Y_{it}=\alpha +\beta_{1}X_{\;\mathrm{it}}+\beta_{2}{\left(X_{it}\right)}^{2}+\beta_{3}M_{it}+\beta_{4}X_{it}\times M_{it}+\beta_{5}{\left(X_{it}\right)}^{2}\times M_{it}+\beta_{k}Z_{k\;it}+\mu_{i}+\delta_{i}+\epsilon_{it}$$
where $M_{it}$ is the moderator, representing the income inequality (Theil) of the provide i in year t. $X_{it}\times M_{it}$, and ${\left(X_{it}\right)}^{2}\times M_{it}$ is the “interaction term”, and we are interested in their coefficient $\beta_{4}$ and $\beta_{5}$. If both $\beta_{4}$ and $\beta_{5}$ are significant, then income inequality has a moderating effect on the “non-linear relationship between economic growth and food safety risk”.

#### 3.3.3. SYS-GMM

To avoid estimate bias caused by potential endogeneity, we introduce a dynamic model lag term. And using the system GMM model to estimate Equations (2) and (3). The model is shown in Equations (4) and (5):(4)$$Y_{it}=\alpha +\overset{m}{\underset{j=1}{\sum }}\gamma_{0}Y_{\;i,t-j}+\overset{n}{\underset{q=0}{\sum }}\gamma_{1}Y_{\;i,t-q}+\overset{n}{\underset{q=0}{\sum }}\gamma_{2}{\left(X_{i,t-q}\right)}^{2}+\overset{c}{\underset{l=0}{\sum }}\gamma_{k}Z_{k\;i,t-l}+\mu_{i}+\delta_{i}+\epsilon_{it}$$
(5)$$Y_{it}=\alpha +\overset{m}{\underset{j=1}{\sum }}\gamma_{0}Y_{\;i,t-j}+\overset{n}{\underset{q=0}{\sum }}\gamma_{1}X_{\;i,t-q}+\overset{n}{\underset{q=0}{\sum }}\gamma_{2}{\left(X_{i,t-q}\right)}^{2}+\overset{b}{\underset{p=0}{\sum }}\gamma_{3}M_{i,t-p}+\overset{n,b}{\underset{q=0,p=0}{\sum }}\gamma_{4}X_{\;i,t-q}\times M_{i,t-p}\;+\overset{n,b}{\underset{q=0,p=0}{\sum }}\gamma_{5}{\left(X_{i,t-q}\right)}^{2}\times M_{i,t-p}+\overset{c}{\underset{l=0}{\sum }}\gamma_{k}Z_{k\;i,t-l}+\mu_{i}+\delta_{i}+\epsilon_{it}$$
where $Y_{\;i,t-j}$ is the lagged term of the food safety risk. $X_{\;i,t-q}$ is the lagged term of economic growth. $M_{i,t-p}$ is the lagged term of the Theil. $V_{k\;i,t-l}$ the lagged term of the control variable. $\gamma_{1}$, $\gamma_{2}$, $\gamma_{3},\;\;\gamma_{4}$ and $\gamma_{5}$ is the corresponding coefficient in which we are interested. And the meanings of other variables are the same as those in Equations (2) and (3).

## 4. Results

### 4.1. Effect of Economic Growth on Food Safety Risk

#### 4.1.1. Benchmark Results

Table 3 shows the effect of economic growth on food safety risk. According to the result of Column (1), the positive correlation between economic growth and food safety risk is not significant. Equation (2) adds the square term of economic growth to the estimation model in Column (1), and Column (2) presents the estimates for Equation (1). It shows that the primary coefficient is significantly positive, and the quadratic coefficient is significantly negative. There is a non-linear relationship between economic growth and food safety risk, but whether there is an “inverted U-shaped” relationship needs further testing. Following Haans et al. [49], we test the “inverted U-shaped” relationship between economic growth and food safety risk from the following three conditions: First of all, the coefficient of the primary term of economic growth is significantly positive, and the coefficient of the quadratic term is significantly negative. Secondly, when the economic growth is at the minimum (9.68), the slope ($\beta_{1}+2\beta_{2}\times economic\;growth)$ is 2.53, which is greater 0. And when the economic growth is at the maximum (11.85), the slope is −1.73, less than 0. Finally, the turning point of the curve ($-\frac{\beta_{1}}{2\beta_{2}})$ is 10.97, which is within the sample interval. All of these meet the criteria of an “inverted U” relationship. Thus, an “inverted U” relationship exists between economic growth and food safety risk, and H1 is established.

#### 4.1.2. Robustness Test

Endogeneity Test: Column (3) of Table 3 reports the estimates for Equation (4). In the estimation of system GMM, the difference of the disturbance term has a first-order autocorrelation but not a second-order autocorrelation, and the model effectively overcomes the endogeneity problem. The corresponding *p* values of the Sargan and Hense tests are greater than 0.1, so there is no overidentification test in the regression results, and the selection of instrumental variables is reasonable. The regression results are reliable and unbiased. Then the estimation results of Column (3) show that the coefficient of the primary term of economic growth is significantly positive, and the coefficient of the quadratic term is significantly negative. In addition, when the economic growth is at the minimum, the slope is 2.27, greater than 0. When the economic growth is at the maximum, the slope is −1.20, less than 0. And the turning point of the curve is 11.10, which is within the sample interval. Consistent with the results of Equation (1), there is an “inverted U-shaped” effect of economic growth on food safety risk.Substitution of dependent variable: We use the logarithm of the number of foodborne disease patients as a proxy for food safety risk and substitute it into Equation (1). The estimation results are shown in Column (4) of Table 3, which is consistent with the estimation results in Column (2). The coefficient of the primary term of economic growth is significantly positive, and the coefficient of the quadratic term is significantly negative. It also meets the three criteria of an “inverted U” relationship, so the “inverted U” relationship between economic growth and food safety risk is robust.Winsorize: The result shown in Column (5) of Table 3 is the estimated result of Equation (1) after all variables after all variables have been subjected to tailoring (1%,99%). This is consistent with the estimates in Column (2) and satisfies the three criteria for the ‘inverted U-shaped’ relationship, so there is a robust ‘inverted U-shaped’ relationship between economic growth and food safety risk.Utest: Table 4 reports the results of the Utest. The results show that the overall t statistic is 1.59, corresponding to a *p* value of 0.056, which was statistically significant at the 10% level. And the Slope contains both positive and negative values. The “inverted U-shaped” impact of economic growth on food safety risk has been confirmed. The “inverted U-shaped” relationship between economic growth and food safety risk is robust.

The above tests show an “inverted U-shaped” relationship between economic growth and food safety risk. When the level of economic growth does not reach the turning point, the food safety risk increases with economic growth. When economic growth crosses the turning point, the food safety risk decreases with increased economic growth, and H1 is established.

### 4.2. Moderating Effect of Income Inequality

According to the estimation results in Table 3, there is a positive correlation between income inequality and food safety risk, but the statistical significance is not robust., so income inequality does not necessarily directly affect food safety risk.

Table 5 shows the test results of the moderating effect of income inequality on the “inverted U-shaped” relationship between economic growth and food safety risk. The estimation result of Equation (3) is reported in Column (1). As for the main effect of economic growth, the primary coefficient of economic growth ($\beta_{1}$) is significantly positive, and the quadratic coefficient ($\beta_{2})$ is significantly negative, verifying the “inverted U-shaped” relationship between economic growth and food safety risk again. For the moderating effect of income inequality, the interaction coefficient between the primary term of economic growth and Theil ($\beta_{4})$ is significantly negative, while the interaction coefficient between the quadratic term of economic growth and Theil ($\beta_{5})$ is significantly positive. It indicates that income inequality has a significant moderating effect on the “inverted U-shaped” relationship between economic growth and food safety risk.

Furthermore, following Haans et al. and Jiao et al. [49,50], the moderation of the inverted U-shaped relationship should be analyzed from two aspects: the shift of the turning point of the curve and the change of the curve shape.

First and foremost, the influence of the moderator on the curve turning point: if the partial derivative of the curve turning point to the moderator is greater than 0, the curve turning point will move to the right under the moderating effect; if the partial derivative of the curve inflexion point to the moderator is less than 0, the curve inflection point will move to the left under the moderating effect. Equation (6) is the curve turning point equation derived from Equation (3), and Equation (7) is obtained by obtaining the partial derivative of Theil from Equation (6). Following the estimates in Column (1) of Table 3, $2{\left(\beta_{2}+\beta_{5}Theil\right)}^{2}$ is always greater than 0; and the value of $\left(\beta_{1}\beta_{5}-\beta_{2}\beta_{4}\right)$ is 0.33, greater than 0. The partial derivative of the turning point of the curve to Theil is greater than 0. Therefore, the turning point of the curve increases with the increase of Theil, and the turning point moves to the right under the moderation of income inequality. H2a is established.
(6)$${\left(Economic\;growth\right)}^{*}=-\frac{\beta_{1}+\beta_{4}Theil}{2\left(\beta_{2}+\beta_{5}Theil\right)}$$
(7)$$\frac{\partial {\left(Economic\;growth\right)}^{*}}{\partial Theil}=\frac{\beta_{1}\beta_{5}-\beta_{2}\beta_{4}}{2{\left(\beta_{2}+\beta_{5}Theil\right)}^{2}}$$

Besides, the influence of the moderator on the curve shape: For Equation (3), the effect of the moderator on the shape of the curve depends on the size and significance of the interaction coefficient between the quadratic term of economic growth and Theil ($\beta_{5})$. If $\beta_{5}$ is significantly positive, the inverted U-shaped curve is flattened; if $\beta_{5}$ is significantly negative, the inverted U-shaped curve is steepened. Following the estimates in Column (1) of Table 3, $\beta_{5}$ is significantly positive. This indicates that under the moderating effect of income inequality, the inverted U-shaped relationship between economic growth and food safety risk is flattened. H2b is verified.

#### Robustness Test


Endogeneity Test: Column (2) of Table 5 reports the results of the estimation of Equation (6). The values of AR indicate that there is an endogeneity problem, but the system GMM model effectively overcomes the endogeneity problem. Sargan test and Hense test show that there is no over-identification test in the model, and the selection of instrumental variables is reasonable. And the estimation result is consistent with Column (1). The first-order coefficient of economic growth ($\gamma_{1})$ is significantly positive; the second-order coefficient ($\gamma_{2})$ is significantly negative; the interaction coefficient of the primary term and Theil ($\gamma_{4}$) is significantly negative; and the interaction coefficient of the quadratic term and Theil ($\gamma_{5}$) is significantly positive. And the value of $\left(\gamma_{1}\gamma_{5}-\gamma_{2}\gamma_{4}\right)$ is greater than 0. In addition, we also use the instrumental variable method. Following Li and Qi [51], we take economic growth with a lag period as the instrumental variable. The estimation result is shown in Column (3) of Table 5, which is also consistent with the estimation results in Column (1). And the results of the endogeneity test in Column (3) show that there is an endogeneity problem. The results of a weak instrumental variables (weak IV) test and an under-identification test show no weak IV or under-identification in our analysis. These endogeneity test results show that income inequality has a significant moderating effect on the inverted U-shaped relationship between economic growth and food security. Under the moderation of income inequality, the turning point of the inverted U-shaped curve between economic growth and food security shifts to the right, and the curve relationship becomes gentler.Substitution of the moderator: In Table 6, Columns (1) and (2) report estimates based on different measures of income inequality. $\beta_{1}$ and $\beta_{5}$ is significantly positive; $\beta_{2}$ and $\beta_{4}$ is significantly negative, and the value of $\left(\beta_{1}\beta_{5}-\beta_{2}\beta_{4}\right)$ is greater than 0, which is consistent with the results in column (1) of Table 5. Therefore, income inequality has a robust moderating effect on the inverted U-shaped relationship between economic growth and food safety risk.Centralized processing of panel data: Following Balli and Srensen [52], we subtract the province-specific average from the economic growth and income inequality, respectively, and then use Equation (3) to estimate it to reflect the difference of slope in different regions, which is more appropriate to the actual situation of each provincial-level administrative regions of China. The estimation results are reported in columns (3) and (4) of Table 6, which are consistent with the estimation results in column (1) of Table 5. “Under the moderation of income inequality, the turning point of the inverted U-shaped curve between economic growth and food safety risk moves to the right, and the curve relationship is flattened.” This conclusion holds in all provincial-level administrative regions in China, and the moderating effect of income inequality is robust.Image description: In Figure 2, when the level of income inequality is low, the inverted U-shaped curve of economic growth and food safety risk reaches the turning point relatively early, and the regions in the high economic growth group have already crossed the turning point and entered the decline phase; the shape of the curve is also steeper. However, when income inequality is high, the inverted U-shaped relationship between economic growth and food safety risk has not yet reached the turning point and is still in the upward phase; the shape of the curve is also flatter. This means that the relationship between economic growth and food safety risk changes with income inequality, and reducing income inequality can help to accelerate the crossing of the inflection point and facilitate the management of food safety risk. All of these show that income inequality has a moderating effect on the inverted U-shaped relationship between economic growth and food safety risk. The increase of income inequality will weaken the sensitivity of food safety risk to changes in economic growth. Under the moderating effect of income inequality, the turning point of the inverted U-shaped curve of economic growth and food safety risk shifts to the right, and the curve shape is flattened. H2, H2a and H2b are established.


## 5. Discussion

To ensure the study’s objectivity, efficacy, and robustness, the data we used were obtained from authoritative organizations such as the National Bureau of Statistics of China and the National Health and Family Planning Commission, and the variables such as food safety risk and economic growth used highly representative indicators. The data are authoritative, representative, and objective. Moreover, the modeling methods we use have passed the Hausman test and the robustness test. Our estimation results are valid and robust.

Firstly, our results show that in China, the food safety risk is a Kuznets curve, and the turning point of the curve is around RMB 58,104.59 per capita GDP (based on prices in 2011), which is greater than the estimated results of Zhang et al. and Yin et al. [11,12]. This difference is due to the fact that Zhang et al. only used China’s time series data and did not consider other control variables when estimating the turning point of the curve [11], which is inaccurate compared with our study using provincial panel data and considering control variables. The turning point of the curve calculated by Yin et al. was the per capita GDP discounted at 2005 base period prices [12], so the turning point difference between us is due to the inconsistent base period, and it is affected by inflation. As of 2020, 13 provinces in China have crossed the turning point, and the overall level of national food safety risk has also crossed the turning point, starting to enter a declining stage. This is consistent with the estimated results of Zhang et al. and Yin et al. [11,12], and is also in line with the basic trend of the “stable and positive” overall situation of food safety in China.

Secondly, the results also show that income inequality has a moderating effect on the inverted U-shaped relationship between economic growth and food safety risk. Under the moderation of income inequality, the turning point of the food safety risk Kuznets curve shifts to the right, and the curve becomes flatter as a whole. This means that income inequality increases the food safety risk, reduces the positive effect of economic growth on the food safety risk, and delays the emergence of the turning point of the curve, which is consistent with the previous theoretical hypothesis and similar to the views of the Tang and Easterlin Paradox.

However, there are still several limitations in our study. First of all, limited by data availability, the time dimension of our research is short and mainly focused on the macro level. In the future, we can expand the time range of investigation through data mining technology and study it at the micro level through surveys to better serve the public and ensure the food safety of residents. Secondly, our research is limited to China, but food safety is a global problem. In the future, we can expand the scope of study to all countries and dig deeply into the differences in food safety in different countries. Thirdly, the factors affecting food safety risk are numerous and complex, but our research mainly focuses on socio-economic and climate factors. In the future, when the problem of data availability is solved, we can introduce more environmental, enterprise and individual characteristics factors to explore the food safety risk and its “turning points” in a more comprehensive way, especially the influence of objective and subjective factors of enterprises and individuals. In addition, the internal logic of how economic growth affects food safety risk and the moderating effect of income inequality needs to be further studied and empirically tested.

In conclusion, our study validates the Kuznets hypothesis of food safety risk and the moderating effect of income inequality. It reminds us that we should consider the factors of the inflection point of the Kuznets curve of food safety risk when managing food safety risk. We should consider positive and negative influencing factors and adopt positive governance measures, laws, and regulations to accelerate the crossing of the inflection point of the Kuznets curve and enter the stage of declining food safety risk in advance.

## 6. Conclusions

Based on China’s provincial panel data from 2011 to 2020, this paper uses the two-way fixed effect model and moderating effect model to study the impact of economic growth on food safety risk and the moderating effect of income inequality. The results show an “inverted U-shaped” relationship between economic growth and food safety risk, and the turning point is about RMB 58,104.59 per capita GDP (based on the price in 2011). China’s food safety risk has crossed a turning point and is declining, but the overall level is still in a high-risk state. In addition, income inequality moderates the relationship between economic growth and food safety risk, making the turning point of the food safety risk Kuznets curve move to the right, and the curve becomes flatter. Income inequality has a significant moderating effect on the food safety risk Kuznets curve. This study can provide the following suggestions for Chinese policymakers.

First, we should deepen the Supply-side Structural Reform to provide the market with high-quality food, thereby reducing the risk of social food safety. Supply-side Structural Reform can help alleviate the gap between supply and demand and provide high-quality and safe food by improving the adaptability and flexibility of supply to changes in demand. At the same time, it can promote scientific and technological innovation and strengthen the research on key technologies of the food industry, which improve the safety of the food supply chain. Moreover, it can promote structural adjustment, optimize the allocation of resources, and improves total factor productivity to stimulate economic growth and reduce the level of food safety risks.

Second, we need to coordinate the development of a green economy and digital economy and improve source pollution of food production and environmental pollution while promoting economic growth, which can jointly promote the improvement of food safety. We should replace the traditional production and consumption mode of “high input, high consumption and high pollution” with an” efficient, harmonious, and sustainable” economic growth mode by developing green industries such as ecological agriculture and recycling industry, which reduces the input of chemicals and industrial pollution in agricultural production to effectively reduce the risk of contamination in the food supply chain and reduce the risk of food safety. In addition, the digital economy is a key force for economic growth. Developing and popularizing the digital economy throughout the country will not only help to promote economic growth but also reduce information asymmetry rapidly. It will accelerate the realization of the inflection point of the “inverted U-shaped” curve of economic growth and food safety risk in various regions.

Third, by constructing a coordinated system of primary distribution, redistribution, and tertiary distribution, we can promote common prosperity and alleviate the negative impact of income inequality. First of all, through the implementation of the policy of education for the benefit of the people and the incentive policy of skill training, the human capital level of the low and middle-income groups can be improved so that they can better adapt to the market economy and increasing the proportion of income in the initial distribution. Secondly, combined with the improvement of the long-term mechanism of reasonable wage growth, the proportion of labor remuneration in the initial distribution should be increased to adapt to price changes to gradually improve living standards and reduce the food safety risk. Finally, by enhancing the redistribution system and the incentive policy of three distributions, we can narrow the income and public service gap between urban and rural areas, within and between regions.

## Figures and Tables

**Figure 1 foods-12-03066-f001:**
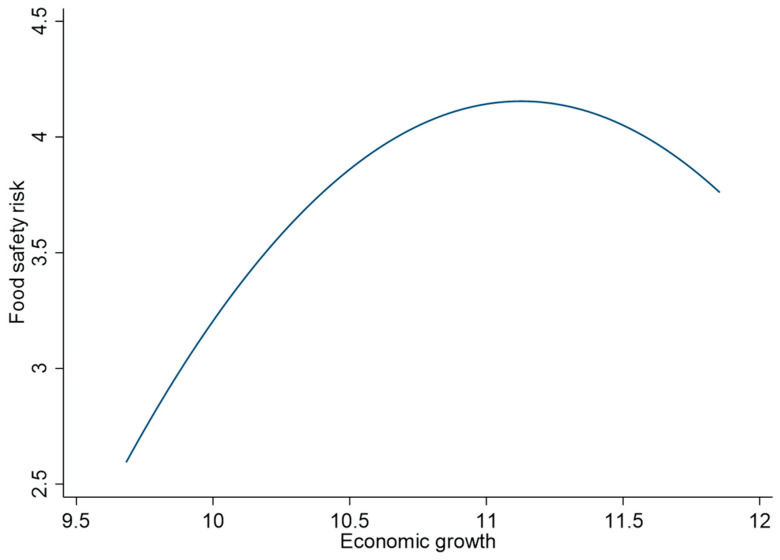
Fitting line between economic growth and food safety risk.

**Figure 2 foods-12-03066-f002:**
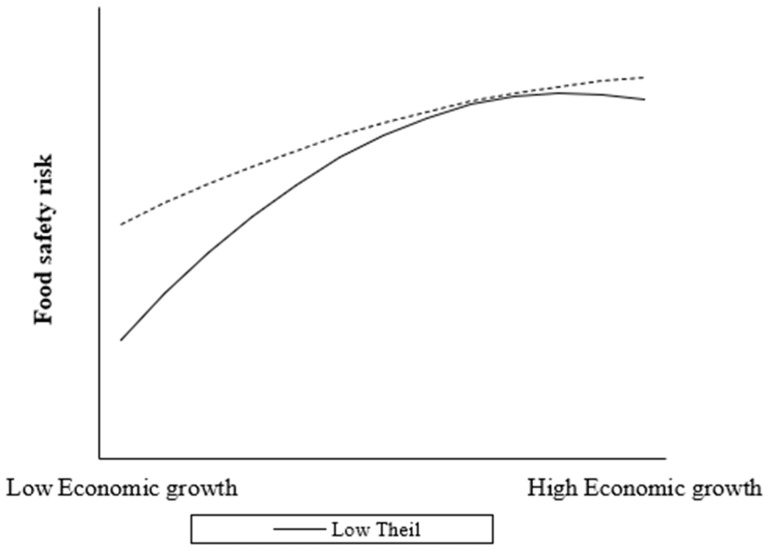
The relationship between economic growth and food safety risk under different levels of income inequality. Based on the results of the curves moderating effect, we plotted the moderating effect of economic growth and income inequality into high and low groups by the mean plus or minus one standard deviation, respectively. The solid curve represents the relationship between economic growth and food safety risk at low-income inequality (Low Theil), and the dotted curve represents the relationship between economic growth and food safety risk at the level of high-income inequality (High Theil).

**Table 1 foods-12-03066-t001:** Description and definition of variables.

Variable	Definition	Observations	Mean	S.D.
Food safety risk	Ln (1 + number of foodborne diseases events)	300	3.93	1.33
Economic growth	Ln (1 + per capita GDP (unit: yuan per person)	300	10.77	0.42
Theil	Calculated by Equation (1)	300	8.87	3.89
Government regulation	Ln (1 + number of personnel in health supervision institutes)	300	7.60	0.70
Industrial structure	Output value of tertiary industry/Output value of secondary industry	300	1.32	0.73
FIV	Ln (1 + Food enterprises output value above scale (unit: 100 million yuan)	300	7.70	1.14
AFTV	Ln (1 + Total output value of agriculture, forestry, animal husbandry and fishery (unit: 100 million yuan)	300	7.80	0.99
Education level	The average schooling of people above six years old (unit: years)	300	9.21	0.89
CPI	Consumer price index (unit: %)	300	102.50	1.18
Temperature	Ln (1 + average temperature of each province (unit: °C)	300	2.63	0.43
Rainfalls	Ln (1 + average rainfall of each province (unit: 0.1 mm^2^)	300	6.78	0.50

**Table 2 foods-12-03066-t002:** Descriptive statistics at different levels of economic growth and income inequality.

		Low-Theil	High-Theil
Food safety risk Average number of Ln (1 + number of food-borne diseases events)	Low Economic growth	2.58	3.58
	Medium Economic growth	4.20	3.82
	High Economic growth	4.21	4.72

**Table 3 foods-12-03066-t003:** Effect of economic growth on food safety risk.

	(1)	(2)	(3)	(4)	(5)
	FE_1	FE_2	SYS-GMM	FE_3	FE_4
Economic growth	1.39	21.50 **	17.76 **	20.71 *	20.10 *
	(2.05)	(10.25)	(8.46)	(11.72)	(10.15)
(Economic growth)^^2^		−0.98 **	−0.80 **	−0.93 *	−0.88 *
		(0.46)	(0.38)	(0.51)	(0.46)
Theil	0.10	0.16 *	0.02	0.38 ***	0.11
	(0.10)	(0.09)	(0.08)	(0.12)	(0.08)
Government regulation	1.96	2.31	2.60	3.52 *	4.91
	(2.36)	(2.33)	(4.64)	(1.99)	(4.15)
(Government regulation)^^2^	−0.13	−0.16	−0.19	−0.25 *	−0.34
	(0.17)	(0.17)	(0.33)	(0.15)	(0.30)
Industrial structure	−0.76 *	−0.97 ***	−0.34	−0.91 ***	−0.89 ***
	(0.38)	(0.34)	(0.40)	(0.27)	(0.32)
FIV	0.36	0.46 *	0.07	0.49 *	0.47 *
	(0.22)	(0.25)	(0.19)	(0.26)	(0.24)
AFTV	0.00	−0.70	−0.71 *	−0.10	−0.72
	(0.68)	(0.73)	(0.41)	(0.72)	(0.72)
Education level	−0.41	−0.31	−0.45	−0.15	−0.30
	(0.30)	(0.27)	(0.63)	(0.31)	(0.29)
CPI	0.13	0.16	0.11	0.15	0.17
	(0.15)	(0.15)	(0.18)	(0.11)	(0.14)
Temperature	−0.08	−0.59	0.55	−3.03 *	−0.71
	(1.26)	(1.26)	(0.75)	(1.78)	(1.34)
Rainfalls	−0.47	−0.57	0.14	−1.04 **	−0.49
	(0.38)	(0.37)	(0.48)	(0.42)	(0.36)
L.Food safety			0.65 ***		
			(0.18)		
Constant	−28.41	−130.53 **	−110.51 **	−126.72 *	−136.35 **
	(29.24)	(56.04)	(52.44)	(66.51)	(55.84)
Province fixed effect	Yes	Yes	Yes	Yes	Yes
Year fixed effect	Yes	Yes	Yes	Yes	Yes
*AR(1)*			0.002 ***		
*AR(2)*			0.847		
*Sargan test*			0.412		
*Hense test*			0.979		
*Observations*	300	300	270	300	300
*R-squared*	0.708	0.716		0.506	0.720

Notes: (a) Robust standard errors in parentheses. (b) *, ** and *** represent the significance level of 10%, 5% and 1%, respectively. (c) fixed-effect model in Column (1), Column (2), Column (4) and Column (5). Additionally, the corresponding superscript is FE_1, FE_2, FE_3 and FE_4 separately. Column (1) adds only the linear term of economic growth under the condition of controlling other variables; Column (2), the quadratic term of economic growth is added based on Column (1); Column (4) substitutes the logarithm of the number of foodborne disease patients as a substitution variable for food safety risk in Equation (1) for estimation; and in Column (5), all variables are winsorized, and then estimated Equation (1). (d) in Column (3), the SYS-GMM method is employed, and the corresponding superscript is SYS-GMM.

**Table 4 foods-12-03066-t004:** The results of Utest.

	Lower Bound	Upper Bound
Interval	9.68	11.85
Slope	2.16	−1.09
t-value	2.77	−1.59
*p* > t	0.003	0.056
Fieller test (95% confidence interval)	10.89	12.95

**Table 5 foods-12-03066-t005:** Moderating effects of income inequality on the non-linear relationship between economic growth and food safety risk.

	(1)	(2)	(3)
	FE	SYS-GMM	2SLS
Economic growth	36.71 **	33.45 *	36.44 ***
	(17.25)	(16.72)	(13.63)
(Economic growth)^^2^	−1.69 **	−1.59 **	−1.54 ***
	(0.71)	(0.75)	(0.59)
Theil	21.48 **	33.21 ***	25.20 ***
	(9.43)	(3.21)	(8.93)
(Economic growth) * Theil	−4.15 **	−6.40 ***	−4.86 ***
	(1.93)	(0.62)	(1.79)
(Economic growth)^^2^ * Theil	0.20 **	0.31 ***	0.24 ***
	(0.10)	(0.03)	(0.09)
Controls	Yes	Yes	Yes
Province fixed effect	Yes	Yes	Yes
Year fixed effect	Yes	Yes	Yes
Endogeneity test			18.767 ***
Under-identification test			71.505 ***
Weak IV test			111.767
AR(1)		0.006 ***	
AR(2)		0.728	
Sargan test		0.459	
Hense test		1.000	
Observations	300	270	270
R-squared	0.733		0.716

Notes: (a) Robust standard errors in parentheses. (b) *, ** and *** represent the significance level of 10%, 5% and 1%, respectively. (c) Moderating effect model in Column (1), Column (3). In Column (3), we use an instrumental variable method, and the corresponding superscript is 2SLS. (d) In Column (2), we use the system GMM method. (e) All models controlled for time-fixed effect, province-fixed effect, and all other control variables are consistent with Table 3.

**Table 6 foods-12-03066-t006:** Results of robustness test: The moderating effect of income inequality.

	(1)	(2)	(3)	(4)
	SV_1	SV_2	Center_1	Center_2
Economic growth	81.74 **	26.52 **	25.00 **	22.16 *
	(35.72)	(9.91)	(10.01)	(11.07)
(Economic growth)^^2^	−3.74 **	−1.19**	−1.14 **	−1.02 *
	(1.60)	(0.46)	(0.45)	(0.52)
Theil	8.26 *	244.70 **	0.14	−0.29
	(4.18)	(96.74)	(0.09)	(0.24)
(Economic growth) * Theil	−1.51 *	−46.13 **	−10.47 *	−49.72 ***
	(0.77)	(18.18)	(5.41)	(17.27)
(Economic growth)^^2^ * Theil	0.07 *	2.17 **	0.49 *	2.38 ***
	(0.04)	(0.86)	(0.27)	(0.82)
Controls	Yes	Yes	Yes	Yes
Province fixed effect	Yes	Yes	Yes	Yes
Year fixed effect	Yes	Yes	Yes	Yes
Observations	300	300	300	300
R-squared	0.715	0.736	0.724	0.744

Notes: (a) Robust standard errors in parentheses. (b) *, ** and *** represent the significance level of 10%, 5% and 1%, respectively. (c) Moderating effect model in Column (1), Column (2), Column (3) and Column (4). (d) In Column (1), we use the Gini instead of the Theil to measure income inequality. In Column (2), we convert the Theil to a binary indicator using the median of the Theil as the criterion, which takes a value of 1 if the Theil is greater than the median and 0 otherwise. Additionally, the corresponding superscript is SV_1 and SV_2 separately. (e) In Column (3), we estimate Equation (3) after subtracting the province-specific average from the economic growth and the Theil. In Column (4), we estimate Equation (3) after subtracting the province-specific average from the economic growth and the binary indicator of Theil. Additionally, the corresponding superscript is Center_1 and Center_2 separately. (f) All models controlled for time-fixed effect, province-fixed effect, and all other control variables are consistent with Table 3.

## Data Availability

The data used to support the findings of this study can be made available by the corresponding author upon request.

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
