# Peer review of "Economic Growth, Income Inequality and Food Safety Risk"

_foods, 2023, doi:10.3390/foods12163066_

Round 1

Reviewer 1 Report

I will preface this report by stating I am something of a historian of food technologies and food safety in developed economies. I have had a modicum of quantitative training, but no expertise or specific knowledge about the Chinese situation, including data validity. As far as I can tell, the statistical analysis seems to be within the boundaries of normal science in this field, but I will defer to others as to ways in which it could be improved or corrected.

There are some things I like about this paper and others with which I am uncomfortable. This being said, some subjects are probably more difficult to address than others in the Chinese context…

On the plus side, the authors are right to point out that “scholars have different views on the relationship between economic growth and food safety risk, and there is a relative lack of complete and systematic mechanism and theoretical analysis” (p. 2). Perhaps the authors could be more explicit on the fact that economic growth over time is typically a decent proxy for an ever more sophisticated division of labor, including ever more sophisticated food production, processing, packaging, transportation, storage and testing technologies, among others. Sustained economic growth should therefore deliver ever safer and cheaper food to ever wealthier and more sophisticated/educated consumers. Going back to 19th century pioneers such as Heinz or Quaker Oats, one can reasonably infer that, in a relatively free market economy, specific firms (from food processors to supermarkets) grow in large part on the basis of brands that convey things like ever greater food safety to consumers. Poisoning your consumers was never a good way to generate repeat business…

Among the things I like less in the paper is the notion that income inequality necessarily affects food safety negatively. As the authors themselves point out on page 4, “Currently, there are few studies on the impact of income inequality on food safety risk and the relationship between economic growth and food safety risk.” That is perhaps because things are a bit more complicated than they assert.

One problem is that in advanced economies, rich and poor people often consume similar things (e.g., milk) or at worst different grades of things produced in the same locations (e.g., apples of various grades produced in the same orchard; more or less expensive cuts of meat coming from the same animal, etc). It may be that in the case of China people of lesser means have access to food that hasn’t been subjected to the same safety standards as food found in modern supermarkets.

My problem is that this was likely the case BEFORE income inequality increased. In other words, has the diet of people of lesser means in China changed all that much over time in terms of food safety? Is the fact that people of more means now have access to safer food the result of people of lesser means having access to lesser food quality than before? Couldn’t one argue then that greater inequality has resulted in an overall SAFER food system, at least to the extent that newly rich people now have access to better food than before? In this context, can one really say that the “increase of income inequality, as a deep-seated cause of China's food safety problems, will reduce trust and increase indifference”? I am genuinely puzzled.

My key issue with the authors, however, is that they seem to link food safety to the “use of chemical materials such as pesticides, fertilizers and hormones, as well as the opportunistic behavior of food producers, and reduces the government's supervision, thus increasing the risk of food safety at the same level of economic growth.” In my opinion, there is a lot that is wrong with this assessment.

First, food adulteration is as old as food markets (See, among others, Renée Johnson. “Food Fraud and “Economically Motivated Adulteration” of Food and Food Ingredients.” Congressional Research Services, January 10, 2014 https://sgp.fas.org/crs/misc/R43358.pdf) Perhaps the authors should know that we recently celebrated the bi-centenary of F. Accum’s 1820 “Death in a Pot” (See Dayan AD, Dayan JD. Accum and food adulteration: A forgotten bicentennial. Toxicology Research and Application. 2021;5. doi:10.1177/23978473211033034 )

To be honest, I am a bit surprised that food specialists would write that the "fertilizer and pesticide era" and the "era of hormone” have allegedly resulted in a “large number of food safety incidents and serious food safety problems” (p. 3). Indeed, the authors double down on this in the conclusion (p. 15) when they call for the replacement of the “traditional production and consumption mode of "high input, high consumption and high pollution" with an" efficient, harmonious and sustainable" economic growth mode by developing green industries such as ecological agriculture and recycling industry, so as to reduce the use of pesticides, fertilizers and other chemicals in agricultural production and soil erosion, as well as improve environmental pollution.”

The problem with this take is that all the data on food safety tells us otherwise. Like many non-specialists, the authors focus on the pesticide-residue molehill while ignoring the germ mountain. And yet, anyone with some familiarity with the history of the topic knows that the good old days were pretty bad. To list but a couple of short entries:

·       -  CDC. 1999. Achievements in Public Health, 1900-1999: Safer and Healthier Foods https://www.cdc.gov/mmwr/preview/mmwrhtml/mm4840a1.htm

·        - Michelle Jarvie. 2014. “History of food safety in the U.S. – part 1” MSU Extension https://www.canr.msu.edu/news/history_of_food_safety_in_the_us_part_1)

As the author should also know, pesticide residues are typically nowhere to be found in terms of actual deaths by food poisoning. Again, they can look up the relevant section of the CDC website  https://www.cdc.gov/foodborneburden/attribution/index.html or The Cleveland Clinic, among others  https://my.clevelandclinic.org/health/diseases/21167-food-poisoning

Indeed, while the WHO might have gone out of its way to pander to environmentalists,  they still admit that the real food safety issues are not related to pesticide residues, but rather to completely natural things like bacteria, viruses, parasites and prions https://www.who.int/news-room/fact-sheets/detail/food-safety These things were not made worse in recent times, quite the contrary. Paradoxically, richer consumers in advanced economies are often at greater risk because of their insistence of buying local/organic food that, for instance, relies more on manure than synthetic fertilizers.

In light of the above, the results for H1 are not surprising at all. Their analysis of inequality on food risk doesn’t quite make sense to me though. I would also point out that the FAO reference they use is mostly about food security (where inequality does indeed make sense), not so much about food safety. I’ll add that I’m not a fan of measurement without theory.

Other issues: There are a few minor language issues throughout the text. To list a few

·       -  “The risk of food safety” is used at least five times throughout the text. This doesn’t make sense to me. Are the authors talking about “the risk of food poisoning” or perhaps “food safety risk”?

·        - construction of urbanization – I believe “urbanization” would do quite well. “Construction of urbanization” sounds weird.

·        - pp 5 and 13, the authors are using the Theil, not Thiel, index.

A few minor typos, but nothing major.

Reviewer 2 Report

Title: When will the Kuznets Curve Turning Point of China's Food Safety Risk Come: Study Based on the Moderating Effect In-come Inequality

Reviewer comments-

A well-conducted study utilizes a robust model known as the Kuznet Curve for analyzing Food Safety and Risk.

The models will also be employed to assess the impacts of policies, address governance issues, and be tested under various scenario-based conditions.

The developed model ought to be implemented in other developing countries.

I have provided several significant comments that should be considered for the revised submission of the manuscript.

1.       Major comments

I. The manuscript lacks an explanation of food safety risk, its components, and its linkage with economic factors.

II. The Model Kuznets curve needs to be thoroughly explained, highlighting its differences from other non-linear models like SIER (Susceptible-Exposed-Infected-Recovered). The data used in the model should be sufficient to predict the turning point and its efficacy. Factors influencing the turning points and strategies to overcome the risks with associated mitigation policies must be discussed in the manuscript.

III. The manuscript should incorporate relevant literature on food security, vulnerability, and risk from reputable sources.

IV. The discussion section must encompass an evaluation of the model's efficacy and robustness, the importance of the data and its components, a comparison with other models, and its application in other relevant scenarios. Additionally, it should address how the model contributes to resolving major issues related to food safety risks and governance concerns.

V. The Reviewer strongly suggests changing the title of the manuscript.

2.           General comments

        I.            Line 38: Provide the citation properly.

      II.            Line 38-39: Economic development is the process and how it should help in food safety (FS) governance?

    III.            Line 40-41: Positive correlation… then what is the need to write FS risk is the product of economic development. Authors not confident in finding the relations relationship between Economic growth and FS risk. It is also not clear the want form the relationship between FS risk or governance. It is good to explain the FS risk and its components, how it is linked with economic factors, which were completely missing in the MS.

    IV.            In the section 2.1 Economic growth and FS risk hypothesis should be properly citated in the MS.

      V.            The study also represents scenarios based FS and risks and their turning points , vulnerability, and their mitigation linked with other developing countries.

    VI.            Section 4.1.2 Robustness test for the model, it should be minimised and moved to methods section and useful for other model comparisons, and discussions.

  VII.            At many of the sentences are not linked with each other and repetition of the sentences and English are not up to the marks.

At many of the sentences are not linked with each other and the repetition of the sentences and English are not up to the mark.
